# Interactions among *Escovopsis*, Antagonistic Microfungi Associated with the Fungus-Growing Ant Symbiosis

**DOI:** 10.3390/jof7121007

**Published:** 2021-11-25

**Authors:** Yuliana Christopher, Celestino Aguilar, Dumas Gálvez, William T. Wcislo, Nicole M. Gerardo, Hermógenes Fernández-Marín

**Affiliations:** 1Centro de Biodiversidad y Descubrimiento de Drogas, Instituto de Investigaciones Científicas y Servicios de Alta Tecnología (INDICASAT AIP), Panamá 0843-01103, Panama; yulianachristopher83@gmail.com (Y.C.); aguilar2587c@gmail.com (C.A.); 2Department of Biotechnology, Acharya Nagarjuna University, Guntur 522510, Andhra Pradesh, India; 3Departamento de Microbiología Médica, Universidad de Panamá, Panamá 0824, Panama; 4Programa Centroamericano de Entomología, Universidad de Panamá, Panamá 0824, Panama; galvezd@gmail.com; 5Smithsonian Tropical Research Institute, Balboa, Ancón, Panamá P.O. Box 0843-03092, Panama; wcislow@si.edu; 6Sistema Nacional de Investigación, Panamá 0816-02852, Panama; 7Department of Biology, Emory University, Atlanta, GA 30322, USA; ngerard@emory.edu

**Keywords:** vegetative incompatibility, intermingling, antagonism, fungus-growing ants, coinfection, Attini, *Escovopsis*

## Abstract

Fungi in the genus *Escovopsis* (Ascomycota: Hypocreales) are prevalent associates of the complex symbiosis between fungus-growing ants (Tribe Attini), the ants’ cultivated basidiomycete fungi and a consortium of both beneficial and harmful microbes found within the ants’ garden communities. Some *Escovopsis* spp. have been shown to attack the ants’ cultivated fungi, and co-infections by multiple *Escovopsis* spp. are common in gardens in nature. Yet, little is known about how *Escovopsis* strains impact each other. Since microbe–microbe interactions play a central role in microbial ecology and evolution, we conducted experiments to assay the types of interactions that govern *Escovopsis*–*Escovopsis* relationships. We isolated *Escovopsis* strains from the gardens of 10 attine ant genera representing basal (lower) and derived groups in the attine ant phylogeny. We conducted in vitro experiments to determine the outcome of both intraclonal and interclonal *Escovopsis* confrontations. When paired with self (intraclonal interactions), *Escovopsis* isolated from lower attine colonies exhibited antagonistic (inhibitory) responses, while strains isolated from derived attine colonies exhibited neutral or mutualistic interactions, leading to a clear phylogenetic pattern of interaction outcome. Interclonal interactions were more varied, exhibiting less phylogenetic signal. These results can serve as the basis for future studies on the costs and benefits of *Escovopsis* coinfection, and on the genetic and chemical mechanisms that regulate the compatibility and incompatibility observed here.

## 1. Introduction

*Escovopsis* (Ascomycota: Hypocreales) is a diverse genus of microfungi that has only ever been found in association with gardens of fungus-growing ants (Tribe Attini). *Escovopsis* spp. are best known as mycoparasites of the basidiomycete fungi that the ants cultivate for food [1,2,3], though the virulence of species varies and the ecological roles of many *Escovopsis* spp. have not been characterized [1]. Co-cladogenesis analyses suggest a pattern indicative of co-diversification of the mutualistic hosts (ants and their cultivated fungi) and *Escovopsis* [4,5,6,7,8,9] at broad taxonomic scales, with less fidelity at finer scales of resolution [10,11,12]. Additionally, a fourth symbiotic taxon, actinomycete bacteria, inhabit the ants’ cuticles and produce antifungal compounds that inhibit the growth of some *Escovopsis* spp. [8,13,14,15,16]. Studies of interactions among members of the system highlight that there can be both competition [17] and coexistence between ant species [18], somatic incompatibility between cultivar strains [19] and antagonistic interactions between actinomycete bacteria [20,21]. Less, however, is known about interactions among *Escovopsis* species or strains. The few studies that have been carried out have been restricted to studies on *Escovopsis* associated to leaf-cutter ants. Their findings indicated little antagonism between the *Escovopsis* spp., which could facilitate coinfection [12,22] and coexistence, however this may not be representative of *Escovopsis*’s diversity. *Escovopsis* coinfections in attine ant gardens have been reported to be found in the colonies of many fungus-growing ant species [12,22,23,24]. Coinfections of a single host by multiple parasites are common in many natural host–parasite systems [25]. Parasite coinfection may facilitate parasites’ abilities to establish, replicate and persist within hosts [26,27]. Alternatively, parasites and pathogens may inhibit one another’s establishment and replication, lessening virulence [28,29,30,31,32,33,34]. Co-infections may also influence transmission processes [35,36], and more broadly the structure and evolution of microbial communities [37,38]. To begin to elucidate how *Escovopsis* microfungi interact with one another, we extensively collected and sequenced *Escovopsis* isolated from colonies of 10 genera of attine ants found in Panama. We randomly selected one strain of *Escovopsis* isolated from gardens of 12 species of ants for use in in vitro confrontations and used a phylogenetic approach to analyze the distribution of interaction outcomes.

## 2. Materials and Methods

### 2.1. Nest Collection and Escovopsis Isolation

We collected colonies from 2013 to 2016 at multiple localities in the Republic of Panama (Appendix A). Small attine ant colonies were maintained in sterile Petri dishes (100 × 15 mm), while larger colonies were placed in plastic containers, previously sterilized with 70% ethanol. We fed colonies corn meal or fresh leaves and placed a piece of paper soaked with sterile water in each container. To isolate *Escovopsis* strains, we followed methods described in [24].

### 2.2. DNA Extraction, Sequencing and Phylogenetic Analyses

DNA from 240 *Escovopsis*-like strains was extracted from mycelia of pure cultures following [39]. We amplified PCR products of two different nuclear DNA markers: the Internal Transcribed Spacer (ITS) and the Large Subunit rRNA (LSU). The primers and conditions used for amplification are described in Table 1. PCR products were cleaned with the QIAquick PCR Purification kit (Qiagen, Hamburg, Germany) following the supplier’s instructions and sequenced by Eurofins Genomics (Germany). DNA sequences were checked and assembled using Geneious Prime 2019 (https://www.geneious.com, accessed on 15 August 2021). All newly generated sequences were deposited to the National Center for Biotechnology Information (NCBI) Taxonomy Database (https://www.ncbi.nlm.nih.gov, accessed on 10 August 2021) GenBank with accession numbers MZ959192-MZ959282 and MZ964338- MZ964401 for LSU and ITS sequences, respectively (Appendix A).

For phylogenetic analyses, we included all strains that were successfully sequenced as well as representative sequences from ten named *Escovopsis* species [39,41,42,43] and one species from the genus *Escovopsioides* [39] obtained from GenBank, as shown in Appendix A. We used *Hypocrea lutea* and *Hypomyces spp.* as outgroups. The final dataset comprised two files, one for LSU with 109 taxa and the other for ITS with 78 taxa.

We performed separate phylogenetic reconstructions for LSU and ITS. Alignments were performed with MAFFT 1.4.0 [44] as implemented in Geneious Prime 2019 (https://www.geneious.com, accessed on 15 August 2021) with default parameters. The appropriate nucleotide substitution model for each gene was determined by using JModelTest2 on XSEDE 2.1.6 [45]. Using AIC, GTR + I + G was estimated as the best-fit model for LSU, and GTR + G was estimated as the best-fit model for the full ITS dataset.

Maximum-likelihood (ML) and Bayesian Inference (BI) analyses were implemented on the CIPRES Science Gateway portal (https://www.phylo.org, accessed on 22 August 2021) using RAxML-HPC2 on XSEDE 8.2.12 and MrBayes on XSEDE 3.2.7) [46,47], respectively. For ML analyses, the default parameters were used. BI was carried out using the rapid bootstrapping algorithm with the automatic halt option. Bayesian analyses included five parallel runs of 5,000,000 generations, with the stop rule option and a sampling frequency of 1000 generations. The burn-in fraction was set to 0.25, and posterior probabilities (PP) were determined from the remaining trees. The resulting trees were plotted using FigTree v.1.4.2 and edited with Graphic 3.1.

### 2.3. Evaluation of Escovopsis–Escovopsis Interaction Outcomes

#### 2.3.1. Intraclonal Confrontation Bioassays

To conduct confrontation in vitro bioassays of intraclonal pairwise combinations, we chose a single *Escovopsis* strain isolated from each one of 12 attine ant species from genera representing lower and derived groups across the attine phylogeny [48,49]. These included the more basal “lower” attine species *Cyphomyrmex muelleri, C. longiscapus, C.* sp., *Apterostigma auriculatum, Ap. collare and Ap. dentigerum,* and the more derived “higher” attine species *Atta sexdens, At. cephalotes, At. colombica, Mycetomoellerius zeteki, Paratrachymyrmex cornetzi* and *Trachymyrmex sp.10*. For each confrontation bioassay, we cut 6.5 mm agar discs from pure *Escovopsis* strain cultures on Potato Dextrose Agar (PDA). These discs contained approximately 5.0 × 10^6^ conidia. The discs were placed on opposite sides of a 100 mm Petri dish. As controls, each of the 12 *Escovopsis* cultures was also paired with a mycelium-free PDA agar block (Appendix A). Each intraclonal interaction and their controls were replicated three times. Petri dishes were maintained in a plastic container at room temperature under a 10:14 hr light:dark cycle. Dishes were photographed after 20 days. We measured the area of growth of all isolates using ImageJ [50]. From photographs, we categorized the outcome of each interaction based on previous studies for fungi [51,52], which are explained further below.

#### 2.3.2. Interclonal Confrontation Bioassays

To assess the outcome of interclonal interactions, we used 8 of the 12 *Escovopsis* strains from the previous experiment. These strains were isolated from gardens of *Apterostigma pilosum*, *Ap*. *collare*, *Cyphomyrmex* sp., *C. longiscapus*, *Trachymyrmex* sp10, *Paratrachymyrmex cornetzi*, *Atta colombica*, and *At. cephalotes*. We used the same protocol as described above. As controls, each of the *Escovopsis* strains were also paired with themselves in intraclonal bioassays and with mycelium-free PDA agar blocks.

#### 2.3.3. Bioassay Outcome Quantification

We categorized the outcome of the interactions as described by [51,52]. The interactions were assigned to one of four outcomes: (1) commensal intermingling (both fungi grow into one another without making an inhibition zone); (2) overgrowth (one isolate grows over the other); (3) invasion/replacement (mycelium of one of the isolates grows into the other and finally replaces it); (4) inhibition, including inhibition at the contact zone (fungi approached each other until almost in contact, with a narrow (1–2 mm) demarcation zone) and inhibition at distance (inhibition at a distance of > 2 mm between fungi) (Figure 1). In addition, we adapted the protocols by [51,52] for analyses based on numerical scores, ranging from one to four, employing three different traits: antagonism index, resistance percentage, and inhibition percentage.

For occurrences of inhibition, we measured the area of the zone of inhibition. The percentage of inhibition (% inhibition) was calculated as the area of an *Escovopsis* strain growing alone (control) divided by the growth area of the same strain in a confrontation bioassay against another *Escovopsis* strain × 100. A value of zero was assigned for those that did not show any inhibition [52]. Moreover, we used the percentage of resistance, which represented the ability of a given species to grow and resist the presence of another fungus, and the antagonism index, which represented the ability of a species to dominate and to compete with other species [52]. To calculate the percentage of resistance and the antagonism index we used the equations by [52].

### 2.4. Statistical Analyses

Based on a phylogeny estimated from ITS sequences of the *Escovopsis* isolates, we analyzed intraclonal confrontation bioassays by using phylogenetically independent contrasts [53] (Garland et al., 1993), as implemented in R (R Development Core Team) [54]) with the function ‘phylANOVA’ (package ‘phytools’), to test for *Escovopsis* clade differences in resistance, area of inhibition (inhibition zone) and antagonism index. We performed 1000 simulations for each test.

For the interclonal confrontation bioassays, to test whether outcomes (type of interaction) varied across combinations of isolates associated with lower attines, derived attines and between the two groups, we performed a multinomial regression as implemented in the function ‘multinom’ (package nnet) and performed Tukey posthoc tests (Tukey-adjusted *p*-values) as implemented in the function ‘lmeans’ (package lsmeans).

## 3. Results

### 3.1. Diversity of Escovopsis Isolates

We collected 220 colonies of 19 species of ten genera of fungus-growing ants: *Atta colombica*, *At*. *cephalotes*, *At*. *sexdens*, *Acromyrmex echinatior*, *Ac*. *octospinosus*, *Trachymyr*mex sp. 10, *Mycetomoellerius zeteki*, *Paratrachymyrmex cornetzi*, *Sericomyrmex amabilis*, *Apterostigma collare*, *Ap*. *dentigerum*, *Ap. auriculatum*, *Ap*. *pilosum*, *Cyphomyrmex longiscapus*, *C*. *muelleri*, *C*. *costatus*, *C*. sp., *Mycocepurus smithii* and *Myrmicocrypta* sp. From these colonies, we isolated pure cultures of 162 *Escovopsis* strains (47 from colonies of *Atta* spp., 16 from colonies of *Acromyrmex* spp., 45 from colonies of *Apterostigma* spp., 12 from colonies of *Cyphomyrmex* spp., 2 from colonies of *Trachymyrmex* sp. 10, 20 from colonies of *Mycetomoellerius zeteki*, 4 from colonies of *Paratrachymyrmex cornetzi,* 6 from colonies of *Myrmicocrypta* sp., 6 from colonies of *Mycocepurus smithii* and 4 from colonies of *Sericomyrmex amabilis*. (Appendix A). Across these samples, we observed that spore color and morphology varied. Brown-spored strains of *Escovopsis* were most commonly isolated from fungus gardens of *Apterostigma*, *Atta*, *Acromyrmex*, *Trachymyrmex*, *Paratrachymyrmex*, *Mycetomoellerius*, and *Sericomyrmex*; pink-spored strains were most commonly isolated from fungus gardens of *Cyphomyrmex*, *Apterostigma* and *Myrmicocrypta*; and yellow-spored strains were isolated from *Apterostigma* and *Mycocepurus* gardens.

A total of 155 new sequences from *Escovopsis* spp. were deposited in the NCBI database and their accession numbers are provided in Appendix A. The alignments of ITS and LSU had lengths of 563 bp and 541 bp, respectively. For both gene markers, the topologies of the BI and ML trees are basically congruent with variable support values, therefore only one tree topology (with nodal support from both methods) is shown for each (Figure 1 and Figure 2). The recently collected *Escovopsis* strains reported in this study were largely grouped according to the colors of the spores (brown, pink and yellow, including yellowish) in both phylogenies (Figure 1 and Figure 2). The *Escovopsis* isolated from colonies of *Atta*, *Acromyrmex*, *Trachymyrmex*, *Paratrachymyrmex*, *Mycetomoellerius* and *Sericomyrmex* grouped together with the named species *E*. *weberi*, *E*. *microspora*, *E*. *moelleri* and *E*. *aspergilloides, E. lentecrescens, E. primorosea, E. catenulata and E. longivesica* (Figure 1 and Figure 2). The brown-spored strains of *Escovopsis* isolated from colonies of *Ap*. *auriculatum*, *Ap*. *dentigerum*, *Ap*. *collare*, *Ap*. *pilosum* grouped together with *E*. *clavatus* and *E. multiformis*. The pink-spored strains of *Escovopsis* isolated from *Ap. collare*, *Ap*. *auriculatum*, *Cyphomyrmex* spp. and *Myrm. ednaella* colonies grouped with *E*. *kreiselli*, the only pink-spored species described for *Escovopsis* (Figure 1 and Figure 2). Most yellow-spored *Escovopsis* strains (including yellowish white) collected from colonies of *Ap*. *collare*, *Ap*. *auriculatum*, *Ap*. *dentigerum* and *Ap*. *pilosum* formed a separate clade in the phylogenies (Figure 1 and Figure 2), while the yellow-spored strains isolated from *Mycocepurus smithii* colonies grouped together with *E*. *trichodermoides* (Figure 2).

### 3.2. Intraclonal Confrontation Bioassays

Intraclonal confrontation bioassays between *Escovopsis* strains exhibited distinct outcomes (Figure 2 and Figure 3). The interactions fell into five general classes: intermingling, overgrowth, invasion/replacement, inhibition at touching and inhibition at a distance (Figure 3 and Figure 4F). Outcomes were consistent across replicates.

*Escovopsis* strains isolated from *Apterostigma* colonies exhibited intraclonal inhibition indicative of vegetative incompatibility (Figure 4). Specifically, we found inhibition at contact for the strain isolated from an *A*. *collare* colony, while the strains isolated from *A*. *pilosum* and *A*. *auriculatum* colonies exhibited inhibition at a distance. *Escovopsis* isolated from *Cyphomyrmex* colonies also exhibited inhibition: the strain isolated from a *Cyphomyrmex* sp. colony exhibited inhibition at the contact point, while the strains isolated from *C*. *muelleri* and *C*. *longiscapus* colonies presented inhibition at a distance. *Escovopsis* strains isolated from colonies of the more derived attines most often exhibited intermingling, though *Escovopsis* from *Trachymyrmex* sp10 exhibited inhibition at contact (Figure 4). Overall, *Escovopsis* strains associated with lower attine ants (*Apterostigma*, *Cyphomyrmex*) tended to be more self-antagonistic than those from gardens of the more derived ant genera (*Trachymyrmex*, *Mycetomoellerius*, *Paratrachymyrmex*, *Atta*), as measured by resistance percentage (1105.0 ± 56.4 vs. 173.3 ± 424.4, respectively; phylogenetic independent contrasts: F = 28.4, *p* = 0.001), inhibition percentage (33 ± 16 vs. 2.9 ± 7, respectively; phylogenetic independent contrasts: F = 16.2, *p* = 0.005) and antagonism index (10.5 ± 1.6 vs. 4.0 ± 2.4, respectively; phylogenetic independent contrasts: F = 29.1, *p* = 0.003).

### 3.3. Interclonal Confrontation Bioassays

Overall, there were differences in the outcomes of intraclonal interactions depending on the ant group (lower vs. derived) to which the strains were associated (LR test, X^2^ = 168.6, *p* < 0.001, Figure 3 and Figure 4). Inhibition was significantly more common for the strain combination lower–lower as compared to the combinations of derived–lower (t ratio = 6.67, *p* < 0.001) and derived–derived (t ratio = 9.6, d.f. = 9, *p* < 0.0001). Overgrowth of one strain over the other one was significantly more common in the combination of derived–lower as compared to the derived–derived combination (t ratio = 4.3, d.f. = 9, *p* = 0.005) and tended to be higher than the lower–lower combination, although this difference was not statistically significant (t ratio = −2.6, d.f. = 9, *p* = 0.07). Invasion/replacement was higher for the derived–lower combination as compared to the lower–lower (t ratio = −4.7, d.f. = 9, *p* = 0.003) and the derived–derived combinations (t ratio = 4.7, d.f. = 9, *p* = 0.003). Finally, intermingling only occurred in the derived–derived combination, making it statistically significantly more common in the derived–derived as compared to the lower–lower (t ratio = −14.4, d.f. = 9, *p* < 0.0001) and derived–lower combinations (t ratio = −14.4, d.f. = 9, *p* < 0.0001).

## 4. Discussion

We highlight the phenotypic and genetic diversity of *Escovopsis* at a local scale and begin to assess the potential for antagonistic and mutualistic interactions amongst *Escovopsis* strains. Our results indicate that some of the described species of *Escovopsis*, which have most often been described based on isolates from South America, occur in Panama. Based on sequences of ITS and LSU, some of the 160 *Escovopsis* strains appear to be representatives of eight named species: *E*. *weberi* (the only previously named species reported from Panama [55]), *E*. *microspora*, *E*. *moelleri*, *E. lentecrescens*, *E*. *clavatus*, *E*. *kreiselli, E. multiformis* and *E*. *trichodermoides*. The brown-spored *Escovopsis* strains isolated from *Ap*. *auriculatum*, *Ap*. *dentigerum*, *Ap. collare*, and *Ap*. *pilosum* grouped together with *E*. *clavatus* and *E*. *multiformis*, consistent with the results of Montoya et al., [43], who isolated these species from colonies of *Apterostigma* spp. in Brazil. The yellow-spored strains of *Escovopsis* isolated from colonies of *Mycocepurus smithii* grouped with *E*. *trichodermoides*, associates of *Mycocepurus goeldii* and *Mycetophylax morschi* in Brazil [42,56]. However, the *Escovopsis* strains isolated from *Acromyrmex* spp. in Panama did not group with *Escovopsis* species isolated from *Acromyrmex* spp. in Argentina [57]. Several clades of Panamanian *Escovopsis* had no match to described species (Figure 1 and Figure 2), highlighting the need for future taxonomic work on *Escovopsis*. We likely did not capture the full diversity of *Escovopsis* in the Panamanian isthmus, and more sampling of colonies of widely distributed ant species, such as *Mycocepurus smithii* (see Kellner et al., 2018) [58] and *Sericomyrmex amabilis*, and from genera not included in this study, such as *Myrmicocrypta*, is needed. Filling in those gaps will provide better insight into the biogeographic distribution of *Escovopsis* species.

As for most organisms, recognition of self, of potential mates and of potential antagonists is crucial for fungi [59]. For ascomycetes, vegetative incompatibility (somatic recognition) frequently occurs when two fungal isolates of the same species come into contact [60]. If strains are from different incompatibility groups, they often reject each other, thus limiting genetic exchange, while if they are compatible, they may fuse. Furthermore, ascomycetes may exhibit recognition between species, including pathogen recognition [60]. Below, rather than speculate as to the genetic, physiological and chemical basis of the compatibility and incompatibility observed, which indeed warrant future investigation, we discuss our results within an ecological and evolutionary framework.

Intraclonal interactions of *Escovopsis* strains led to two outcomes, inhibition and intermingling. Intraclonal inhibition was exhibited by *Escovopsis* strains associated with the lower attines (*Apterostigma* spp., *Cyphomyrmex* spp.) and the derived attine *Trachymyrmex* sp. 10, while non-antagonistic intermingling was exhibited by *Escovopsis* strains associated with the derived attines *Mycetomoellerius zeteki*, *Paratrachymyrmex cornetzi* and *Atta* spp. (Figure 2, Figure 3 and Figure 4A–D). Some ancestral characteristics seem to be retained in derived taxa, such as the *Escovopsis* strain isolated from *Trachymyrmex* sp. 10 that exhibited inhibitory interactions between clonal isolates (Figure 2, top of phylogeny). It is possible that interactions are regulated by genetic factors (i.e., compatibility/incompatibility loci) and mediated by the production of secondary compounds with antifungal properties. *Escovopsis* spp. produce an array of secondary metabolites [24,55,61,62], some of which have fungistatic and antifungal properties [61,62]

The ecological relevance of variation in self-compatibility is somewhat unclear. One hypothesis is that recognizing and inhibiting growth of self may be important in smaller colonies, such as those of the lower attines [63], where competing with oneself through vegetative overgrowth or intermingling could be more likely, and thus selected against. Alternatively, vegetative incompatibility may be selected for to prevent the spread of parasites (e.g., viruses), as has been seen in other systems [64,65]. Finally, vegetative incompatibility may be a byproduct of selection for inhibiting non-self competitors. Coinfections of colonies are common [22,24,66] and, due to increased resource competition, coinfection of the smaller, lower attine colonies may place stronger selection on *Escovopsis* associated with these colonies to maintain mechanisms to inhibit the growth of other strains of the same species or other species.

Interclonal, interspecific interactions have been investigated in other systems in the context of how interactions may shape community assembly [52,67]. Results of interclonal interactions between *Escovopsis* ranged from no antagonism (i.e., intermingling) to varying degrees of antagonism (i.e., overgrowth, invasion/replacement, inhibition). Some form of antagonism was observed in 82% of interclonal assays, suggesting that interference competition between *Escovopsis* spp. may be a common consequence of coinfections. Despite this, coinfections are common in nature, which suggests either that the processes shaping *Escovopis*–*Escovopsis* interactions differ in more natural settings or that spatial segregation within gardens allows for co-existence. Future research should explore the dynamics of coinfection in the context of a larger array of *Escovopsis* pairings, employing tractable assays, such as those used here, and infections of ant gardens.

Interference between *Escovopsis* is interesting in light of research highlighting that that *Escovopsis* spp. exhibit host specificity [11,68]. For example, the pink-spored *Escovopsis,* like *E. kresilli*, are found in colonies of lower but not derived attines. *Escovopsis*’ host ranges are shaped by both cultivar defenses [69] and actinomycete bacteria defenses [70,71], as strains of both microbes vary in terms of what *Escovopsis* spp. they inhibit. Interactions with the ants, other microbes and commensals in the gardens, along with additional biotic and abiotic factors, may also shape *Escovopsis* spp. specificity [11]. Our results indicate that interactions with other *Escovopsis*, some of which inhibit each other, should also be taken into consideration.

## 5. Conclusions

To summarize, an extensive collection of *Escovopsis* associated with the breadth of fungus-growing ant species found in Panama reveals unexplored diversity of this genus. While some *Escovopsis* spp. exhibit vegetative incompatibility, others do not. Furthermore, interspecific interactions differ markedly in their outcomes, with many exhibiting signs of antagonism, while others suggest that some *Escovopsis*–*Escovopsis* interactions may be neutral or mutualistic. These interactions should be further explored in terms of both the underlying proximate genetic and chemical mechanisms and how they shape ecological dynamics and evolution in the fungus-growing ant symbiosis.

## Figures and Tables

**Figure 1 jof-07-01007-f001:**
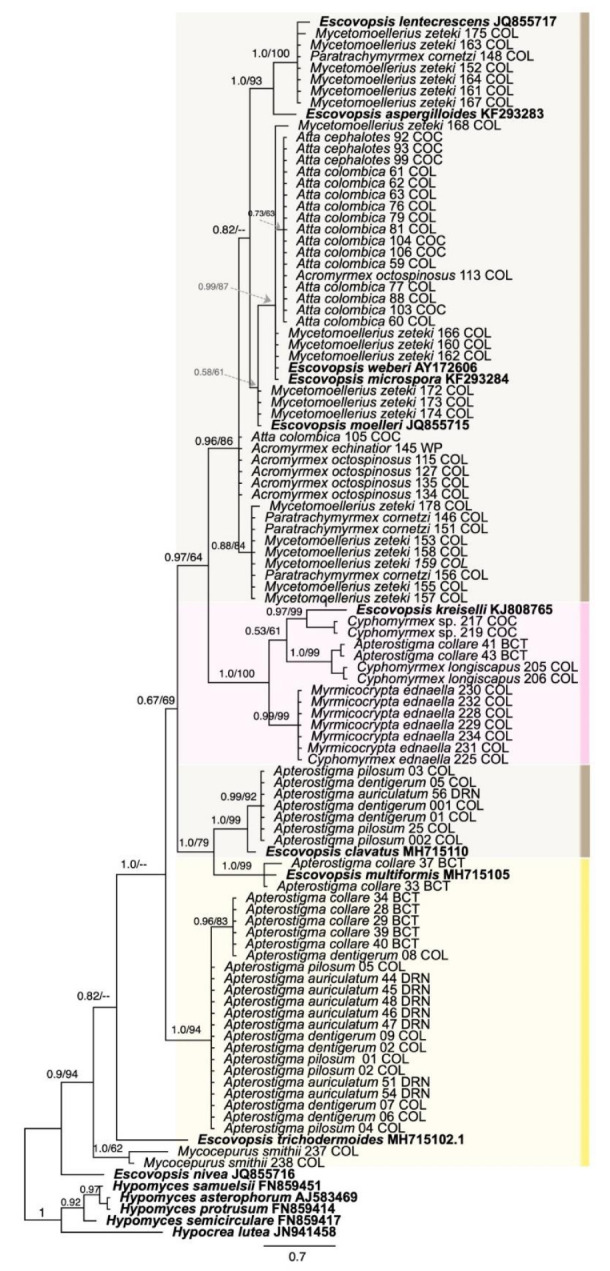
Phylogeny based on Large Subunit rRNA (LSU) of *Escovopsis* strains from fungus-growing ant gardens. The phylogenetic tree shown is based on the Bayesian tree topology. Each fungal strain isolated as part of this study is indicated by the species name of the ant host garden from which the *Escovopsis* was isolated, and all other strains are indicated by their genus and species name, followed by the Genbank accession number. *Hypocrea lutea* was used as an outgroup. Different colors indicate the *Escovopsis* morphotypes shown in the phylogenetic analyses. The numbers on branches indicate the posterior probabilities and the bootstrap support values, respectively. Only bootstrap supports ≥ 50% of ML and the posterior probability values ≥ 0.5 of BI analyses are indicated above or below the respective branches. The provinces from which each strain was isolated are indicated by the following abbreviations: PA, Panamá; COC, Coclé; COL, Colón; DRN, Darién; BCT, Bocas del Toro; WP, West Panamá. Scale bar 0.7 substitutions per site.

**Figure 2 jof-07-01007-f002:**
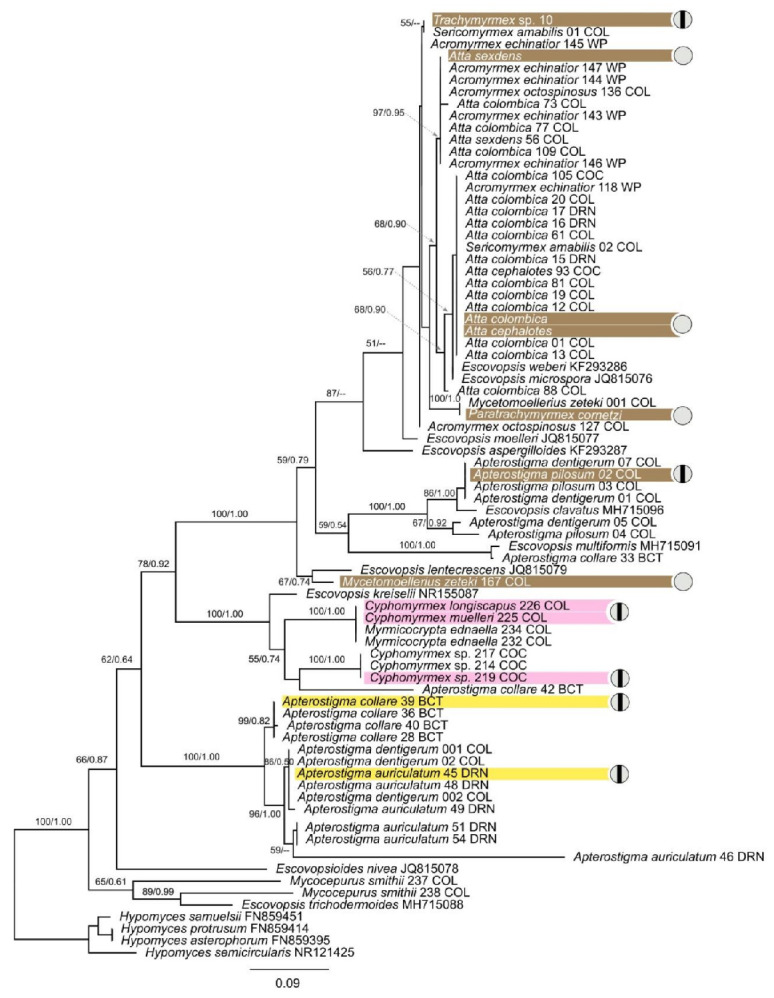
Phylogenetic tree based on ITS rDNA sequences of *Escovopsis* strains from fungus-growing ant gardens. The phylogenetic tree shown is based on the maximum likelihood tree topology. *Escovopsis* strains isolated from lower attine colonies (*Apterostigma auriculatum*, *Ap*. *collare*, *Ap*. *pilosum*, *Cyphomyrmex longiscapus*, *C*. *muelleri* and *Cyphomyrmex* sp.) and strains isolated from derived attine colonies (*Trachymyrmex sp10*, *Mycetomoellerius zeteki*, *Paratrachymyrmex cornetzi*, *Atta colombica*, *At*. *cephalotes* and *A*. *sexdens)*. Each strain isolated as part of this study is indicated by the species name of the ant host garden from which the *Escovopsis* was isolated, and all other strains are indicated by their genus and species name, followed by the Genbank accession number. *Hypomyces* species were used as outgroups. Bootstrap supports (≥50%) of ML and the posterior probability values (≥0.5) of BI analyses are indicated above or below the respective branches. Branch color indicates fungal spore color. The provinces from which each strain was isolated are indicated by the abbreviations detailed in Figure 1. Closed grey circles show intermingling, a non-antagonistic form of interaction, while split grey circles mean inhibition, a form of antagonistic interaction. Scale bar represents 0.09 substitutions per site.

**Figure 3 jof-07-01007-f003:**
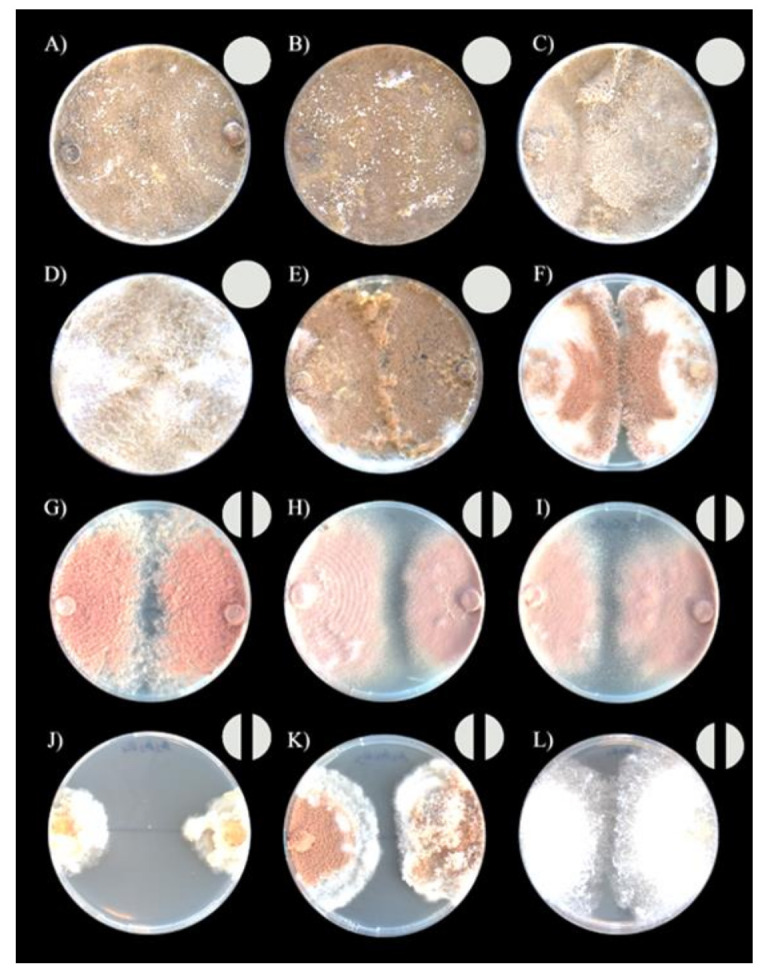
Intraclonal confrontation bioassays between *Escovopsis* strains. (**A**) *Escovopsis* sp. from *At*. *colombica*, intermingling; (**B**) *Escovopsis* sp. from *At*. *cephalotes*, intermingling and (**C**) *At*. *sexdens*, intermingling; (**D**) *Escovopsis* sp. from *Par*. *cornetzi*, intermingling; (**E**) *Escovopsis* sp. from *Myc*. *zeteki*, intermingling; (**F**) *Escovopsis* sp. from *T*. *sp10*, inhibition; (**G**) *Escovopsis* sp. from *C*. sp. colony, inhibition; (**H**) *Escovopsis* sp. from *C*. *longiscapus* colony, inhibition; (**I**) *Escovopsis* sp. from *C*. *muelleri* colony, inhibition; (**J**) *Escovopsis* sp. from *Ap*. *auriculatum* colony, inhibition; (**K**) *Escovopsis* from *Ap. pilosum* colony, inhibition and (**L**) *Escovopsis* sp. from *Ap*. *collare* colony, inhibition. Each photo shows in the top right the symbol of the interaction type that represents it.

**Figure 4 jof-07-01007-f004:**
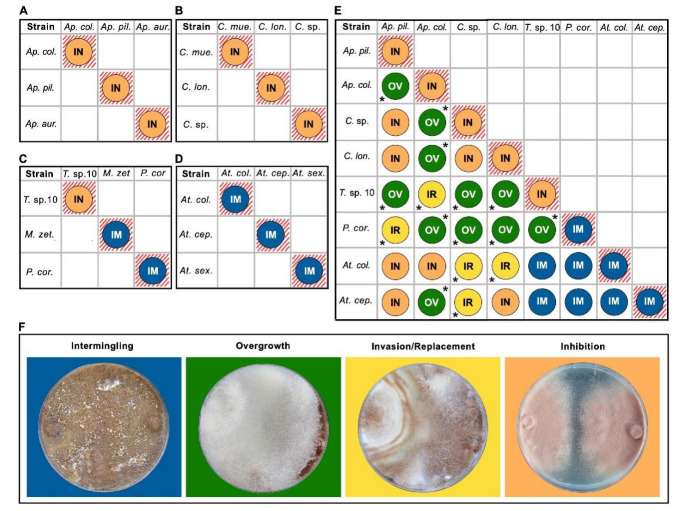
Outcomes of intra- and interclonal confrontation bioassays. The left, top quadrants (**A**– **D**) indicate outcomes of intraclonal interactions for 12 *Escovopsis* strains isolated from colonies of the lower attine ant genera, *Apterostigma* (**A**) and *Cyphomyrmex* (**B**), and derived genera, including *Trachymyrmex*, *Mycetomoellerius*, and *Paratrachymyrmex* (**C**), and *Atta* (**D**). The right, top quadrant (**E**), indicates results of factorial interactions among eight *Escovopsis* isolated from colonies of eight attine ant species. Boxes with lines are intraclonal interactions, and those without lines are interclonal interactions. The asterisk positioned on the bottom left of the interaction type symbol indicates that the strain that grows in excess or replaces it is the one indicated in the row, while an asterisk positioned on the top right of the interaction type indicates that the strain that grows in excess or replaces it is the one indicated in the column. (**F**) Representative interaction outcomes. Intermingling occurs when both fungi grow into one another without any inhibition zone. Overgrowth occurs when one strain grows towards and completely covers the other. Invasion/replacement occurs when one strain grows towards the other and then begins to consume it, or in some cases, completely replaces it. Inhibition occurs when strains approach each other but leave a demarcation line between them. Abbreviations and full names: *Ap. col*, *Apterostigma collare*; *Ap. pil*, *Ap*. *pilosum*; *Ap*. *aur*, *Ap. auriculatum*; *C*. sp, *Cyphomyrmex* sp.; *C*. *mue*, *C. muelleri*; *C*. *lon*, *C. longiscapus*; *At*. *col*, *Atta colombica*; *At*. *cep*, *At. cephalotes*; *At*. *sex*, *At. sexdens*; *T*. sp 10, *Trachymyrmex* sp. 10; *Myc*. *zet*, *Mycetomoellerius zeteki*; *Par*. *cor*, *Paratrachymyrmex cornetzi*.

**Table 1 jof-07-01007-t001:** Primers used for PCR and sequencing of *Escovopsis* strains.

Marker	Primers	PCR and Sequencing Conditions
ITS	ITS4- 5′TCCTCCGCTTATTTGATTATTGATC3′ ITS5- 5′GGATATGTATATATATGTCGTATATCATATGG3′ [40]	3 min of denaturation at 96 °C, 35 cycles consisting of 1 min at 94 °C, 1 min at 55 °C and 2 min at 72 °C [41]
LSU	CLAF-5′GCATTATTCATATTATATGCGGATGGAT 3′ CLAR-5′GATCTCCTTGGTCCGTGTTTCAT 3′ [7]	2 min of denaturation at 95 °C, 40 cycles of 30s at 95 °C, 1 min at 62 °C, 90 s at 72 °C and 5 min of extension at 72 °C [39]

## Data Availability

All sequences generated in this study were submitted to GenBank.

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
