# Peer review of "Interactions among Escovopsis, Antagonistic Microfungi Associated with the Fungus-Growing Ant Symbiosis"

_jof, 2021, doi:10.3390/jof7121007_

Round 1
Reviewer 1 Report
As noted in the paper, previous work on this topic has shown Escovopsis spp. do not antagonize each other and may facilitate coinfection, however, the species diversity used in previous experiments was limited. This study adds to previous work in a clear and compelling way, with broader diversity of Panamanian Escovopsis isolates ranging across the phylogenetic diversity of fungus-growing ants. The author might specifically point out that the previous work focused specifically on leaf-cutter ants in the introduction.
The figures are great for understanding the paper. One thing to consider would be to make it more clear in Figure 2 what the paper uses as species groups. Additionally, a visual legend for the gray circles would be helpful, with representative examples of the interaction types from Figure 3 to help the reader visualize the process. One question I have regarding Figure 4 A-D is if the strains on the columns are the same isolates as the strains on the rows? Or are they different isolates from the same colony? Additionally, for Figure 4E it isn’t clear to me in the case of overgrowth and invasion/replacement if there is an identifiable strain that overgrows or replaces. It might be interesting to include this information.
L45, ‘has’ should be ‘have’ for the plural ‘spp.’
There’s a hanging closed parenthesis on line 51.
Line 170: ‘carried’ should be ‘performed’
Line 309: ‘South American’ should be ‘South America’
Author Response
Dear Editor,
We are grateful to the editor and reviewers for their time and constructive comments on our manuscript. We have implemented their comments and suggestions and wish to submit a revised version of the manuscript for further consideration in the journal.
Changes in the initial version of the manuscript are either highlighted for added sentences or strikethrough for deleted sentences in the revised version. Below, we also provide a point-by-point response explaining how we have addressed each of the editors or reviewer´s comments. We look forward to the outcome of your assessment
Please find attached a point-by-point response to reviewer´s concerns. We hope that you find our responses satisfactory and that the manuscript is now acceptable for publication
Yours sincerely,
On behalf of the co-authors
Hermógenes Fernández-Marín

Reviewer 2 Report
Christopher et al. analyze the phylogenetic relationship of Escovopsis spp. isolated from the gardens of the fungus-growing ants and discuss how Esovopsis strains impact each other by using the inter- and intra-clonal confrontation bioassays. The concept is very interesting in the scope of the symbiosis of fungi and ants. But there are some unclear parts as described below. Please explain them in the revising process.
- L55-57 This sentence seems be unclear. Please rewrite it.
- In the intra- and inter-clonal confrontation bioassays, the photos of the control experiments (L127-128) should be shown in the supporting information. In addition, please explain how the authors selected the representatives of each fungus. For example, the yellowish and brown Escovopsis spp. from the Apterostigma pilosum were separately classified in the phylogenetic tree (Fig. 1). The yellow and pink Escovopsis spp.were also isolated from Apterostigma collare. Why did the authors select the brown (02) and yellow (39) fungi as the representatives for A. pilosum and A. collare, respectively?. The authors should add the comments on the majority of the Escovopsis species in each of the ant gardens (future research?).
- In Fig. 1, please check the sequence of E. longivesica KU298296 and be careful when the authors align the sequences. I think this tree is very strange. If the phylogenetic tree is correct, I would like to see the sequences of KU298290, KU298285, and KU298296 aligned by employing ClastalW. If incorrect, the sequence should be removed.
- In Fig. 2, please make the difference of the circles between “thin” and “thick” clearer.
Author Response

(The authors gave the same response as above.)

Round 2
Reviewer 2 Report
I think the manuscript has been revised properly, but the authors forget to remove the previous Figs 1 and 2 from the revised version.
